# Mass Transport of Dye Solutions through Porous Membrane Containing Tannic Acid/Fe^3+^ Selective Layer

**DOI:** 10.3390/membranes12121216

**Published:** 2022-12-01

**Authors:** Hluf Hailu Kinfu, Md. Mushfequr Rahman, Nicolás Cevallos-Cueva, Volker Abetz

**Affiliations:** 1Helmholtz-Zentrum Hereon, Institute of Membrane Research, Max-Planck-Straße 1, 21502 Geesthacht, Germany; 2Institute of Physical Chemistry, University of Hamburg, Martin-Luther-King-Platz 6, 20146 Hamburg, Germany

**Keywords:** Spiegler–Kedem model, reflection coefficient, solute permeability, tannic acid–metal ion membranes, metal polyphenol complex membranes, dye retention

## Abstract

Tannic acid (TA)–Fe^3+^ membranes have received recent attention due to their sustainable method of fabrication, high water flux and organic solutes rejection performance. In this paper, we present a description of the transport of aqueous solutions of dyes through these membranes using the transport parameters of the Spiegler–Kedem–Katchalsky (SKK) model. The reflection coefficient (*σ*) and solute permeability ($P_{S}$) of the considered TA–Fe^3+^ membranes were estimated from the non-linear model equations to predict the retention of solutes. The coefficients *σ* and $P_{S}$ depended on the porous medium and dye molecular size as well as the charge. The simulated rejections were in good agreement with the experimental findings. The model was further validated at low permeate fluxes as well as at various feed concentrations. Discrepancies between the observed and simulated data were observed at low fluxes and diluted feed solutions due to limitations of the SKK model. This work provides insights into the mass transport mechanism of dye solutions and allows the prediction of dye rejection by the TFC membranes containing a TA–Fe^3+^ selective layer using an SKK model.

## 1. Introduction

Wastewater, especially industrial effluents from textile plants, contains high concentration of dyes. At up to approx. 54% of the total, the textile industry is the main contributor of dye wastewater globally [1]. It is estimated that more than 15% of textile dyes used for dyeing operations ends up in wastewater streams, which is harmful to the environment [2]. Membranes offer a clean, economical and effective method of dyestuff recovery and wastewater treatment [1,3]. Due to their high retention rate, nanofiltration and reverse osmosis have been used for the treatment of industrial wastewater containing dyes [4,5,6,7]. Reverse osmosis is notorious for its high energy consumption and delivers low permeate flux. Nowadays, porous-high-flux membranes, such as ultrafiltration and loose nanofiltration membranes, which operate at low pressure gradients, appear to be more attractive [8,9,10]. Therefore, the development of new porous membranes with a specific focus on the removal of dyes from an aqueous solution is a vibrant research topic [11,12,13,14]. Moreover, efforts have been made to utilize porous membranes for the separation of individual dyes from an aqueous solution containing a mixture of dyes [15]. The development of new membranes and the optimization of membrane-casting parameters to find a good balance between the counteracting permeance and selectivity of the membranes is important [16]. By comparison, the comprehensive understanding of the flux and rejection behavior of a membrane using predictive mass transport models has received significantly less attention. Predictive mass transport models can be utilized to check the feasibility of separating organic solutes (e.g., dye) from a solution with a specific membrane. A good predictive model can provide insight regarding the structural property correlations of membranes and thereby facilitate the optimization of membrane fabrication parameters. Moreover, mass transport models can be expanded to visualize the optimal design of membrane modules and even membrane-based processes to some extent. Several models have been utilized to demonstrate component transport and rejection in porous films. However, there is a lack of understanding regarding the correlations between the nature of solutes, membrane morphology and modes of transport. The availability of empirical methods for the accurate determination of a membrane’s characteristics (e.g., tortuosity and surface charge density) to be used as model input parameters is a major limitation in the progress of mass transport modeling of porous membranes [17].

The Spiegler–Kedem–Katchalsky (SKK) model, a phenomenological approach defined through irreversible thermodynamics, is widely used to analyze the solute retention behavior of nanofiltration membranes. The model was first derived by Kedem and Katchalsky and was then modified by Spiegler and Kedem [18]. Being a phenomenological model, the SKK model does not take into account membrane characteristics. The membrane is considered a black box in this approach. However, the relationship between driving forces and permeate flux can be expressed explicitly [19]. The SKK model correlates the volumetric flux of a two-component (i.e., one solvent and one solute) solution with the flux of the solute. Hence, it provides a way to predict the retention and permeability of the solute in a membrane [20]. The SKK model was first employed for a reverse osmosis membrane. Since then, this phenomenological model has been extensively used to analyze and describe nanofiltration [21] and diafiltration [22] membranes. Nayak et al. [23] successfully employed the SKK model as a tool for predicting the retention of pharmaceutical pollutants from an aqueous solution using nanofiltration membranes. Furthermore, it was demonstrated that the SKK model could successfully predict dye–salt separation performance [24]. This model was further extended for the prediction of dye rejection of both monovalent and divalent charged anionic and cationic dyes [25]. A good agreement of model-calculated and experimental rejection coefficients was confirmed, and this emphasized the significance of the phenomenological model for assessing membrane separation performance.

In this work, for the first time, we examined the validity of the SKK model to predict the retention of dyes using porous thin-film composite (TFC) membranes containing a metal–polyphenol network (MPN) selective layer. In this type of membrane, the selective layer is formed by coordination networks of self-assembled polyphenols, such as tannic acid (TA) with transition metal ions. MPN selective layers are synthesized in a green way using only aqueous solutions. This novel strategy has been used for TA–Ti^4+^ [26], TA–Fe^3+^ [27] and TA–Ni^2+^ [28] selective layer preparation. MPN-containing membranes have been studied for water–oil emulsion, heavy metal removal and nanofiltration applications [26,29,30]. Fang et al. fabricated loose nanofiltration membranes by first blending Fe in PES through NIPS, and then constructing a selective layer via coordination chemistry of TA and Fe^3+^ [31]. The synthesized membrane exhibited excellent dye/salt fractionation with a high rejection to many dyes. Moreover, the introduction of TA improved the membrane hydrophilicity, negative charge and fouling resistance. The optimization of metal–polyphenol coatings and their application for low-pressure filtration is emerging. However, to the best of our knowledge, no model prediction of the component transport and solute rejection performance of TFC membranes with MPN selective layers has been reported so far. Hence, it is necessary to establish a viable method for a predictive approach to assess the solution filtration through these membranes. This study aims to elucidate the mass transport of dye solutions through a TA–Fe^3+^ thin selective layer using the SKK model. Two TFC membranes containing TA–Fe^3+^ selective layers synthesized using aqueous solutions of different pH were examined in this study. The solute transport through the membranes was considered to be a combined effect of diffusive and convective transport in the SKK model. The driving forces of diffusion and convection were the concentration gradient across the membrane and the transmembrane pressure, respectively. Therefore, we studied the dye retention behavior of the prepared TFC membranes using two sets of experiments: (i) using a constant feed concentration and varying the transmembrane pressure, and (ii) using a constant transmembrane pressure with varying feed concentrations.

## 2. Materials and Methods

### 2.1. Chemicals

Tannic acid (TA) was purchased from Sigma-Aldrich Chemie GmbH (Schnelldorf, Germany). Iron salt of iron(III) chloride hexahydrate (FeCl_3_·6H_2_O) was purchased from Alfa Aesar GmbH & Co. (ThermoFisher, Kandel, Germany). In-house prepared polyacrylonitrile (PAN) membranes were used as the support layer of the TFC membranes. Sodium hydroxide (NaOH), orange II (350.32 g/mol), riboflavin (376.36 g/mol) and naphthol green B (878.46 g/mol) were obtained from Sigma-Aldrich. All chemicals were used without modification.

### 2.2. Preparation of Tannic Acid/Fe^3+^ Selective Layer

The concept of supramolecular self-assembly between TA and transition metal ions was used to prepare the selective layers of two thin-film composite membranes. The membranes were synthesized using the sequential deposition of TA and iron salt solutions over a porous PAN support. The top surface of the support was exposed to a 50 mL aqueous solutions of 0.1176 mM of TA and 3.33 mM of FeCl_3_·6H_2_O alternately for 4 min. This molar concentration corresponded to 1 TA: 4.5 FeCl_3_·6H_2_O (1 TA–4.5 Fe in short) in weight ratio. Two layers of TA–Fe^3+^ film were deposited using a layer-by-layer technique. The pH value of the aqueous solution used for dissolving TA was varied. Two membranes, hereby named M1 and M2, were prepared with a pH of 5.8 (deionized (DI) water) and 8.5, respectively. The pH value of 8.5 for the aqueous solution of the M2 membrane was adjusted with a 1 M NaOH solution. Appendix A shows a photograph of the PAN membrane before and after coating.

### 2.3. Membrane Characterization

The membrane morphology was investigated with scanning electron microscopy (SEM). SEM images were recorded on a Merlin SEM (Zeiss, Jena, Germany) at accelerating voltages between 1.5 and 3 keV using an InLens secondary electron detector. Before measurement, the samples were dried under vacuum at 60 °C for 72 h and were sputter-coated with 1–1.5 nm platinum using a CCU-010 coating device (Safematic, Zizers, Switzerland). Cross-fractured specimens were prepared in liquid nitrogen. The pure water flux of M1 and M2 membranes was measured using DI water of 0.055 µScm^−1^. The separation performance of the membranes was analyzed with rejection experiments of three solutes: orange II, naphthol green B and riboflavin. Measurements were performed using a stirred test cell from Millipore (EMD Millipore XFUF07601) with a reduced effective membrane area of 1.77 cm^2^ in a dead-end filtration mode. DI water was passed for 2 h at a transmembrane pressure of 4 bar. Then, retention measurements were performed under two sets of operating conditions: (i) a 0.1 mM feed solution concentration and variable transmembrane pressure in the range 0.2–4 bar, and (ii) a 3 bar transmembrane pressure and variable concentration in the range 0.01 mM–1 mM. The water flux ($J_{w}$) and dye rejection rate (*R*) were calculated using the following equations:(1)$$J_{w}=\frac{W}{\rho *A*t}$$
where $J_{w}$ (kg·m^−2^·s^−1^) is the pure water flux, W (kg) is the weight of the collected permeate, A (m^2^) is the effective membrane filtration area and t (s) is the operation time.
(2)$$R\;\left(\%\right)=\left(1-\frac{C_{p}}{\left(C_{f}+C_{r}\right)/2}\right)*100$$
where R is the solute retention; and $C_{f}$, $C_{p}$ and $C_{r}$ are the concentrations of the feed, permeate and retentate solutions, respectively. Each experimental result in this work was obtained as an average value of at least three replications.

### 2.4. Fitting/Prediction of Dye Rejection with Spiegler–Kedem–Katchalsky Model

Predictions of the separation performance of the membranes were performed using the Spiegler–Kedem–Katchalsky (SKK) model based on irreversible thermodynamics. In the SKK model, the characteristics of the solute (e.g., size and charge) and membrane characteristics (e.g., pore size and surface charge density) are not used as input parameters. For the transport of a two-component solution, i.e., one solvent and one solute, the governing transport equations of the SKK model are as follows:(3)$$J_{v}=L_{p}*\left(\Delta P-\sigma \Delta \pi \right)$$
(4)$$J_{S}=P_{S}*\frac{dC_{s}}{dx}+\left(1-\sigma \right)*C_{S}*J_{v}$$
where $J_{v}$ is the volumetric permeate flux and $J_{S}$ is the solute flux; Δ*P* and Δπ are the applied pressure difference and osmotic pressure difference, respectively; $L_{p}$ and $P_{S}$ are water permeability and solute permeability coefficients, respectively; $C_{s}$ is the solute logarithmic mean concentration between the feed and permeate; and σ is the reflection coefficient of the membrane. The reflection coefficient, σ, describes the semipermeablity of the membrane. It ranges from 0 to 1, with 1 characterizing a perfectly semipermeable film (complete solute ejection). It is also a measure of rejection of the membrane at high flux. It can be observed that Equation (4) contains two parts, the first term depicting diffusion and the second representing convection.

The rejection of a solute is expressed in Equation (2). In the SKK model, the rejection of a solute can be written as follows:(5)$$R=\frac{\sigma \left(1-F\right)}{1-\sigma F}$$
where
(6)$$F=exp\left(-\frac{\left(1-\sigma \right)}{P_{S}}*J_{v}\right)$$

Substituting F into the rejection expression results in a simplified SKK model of the final form:(7)$$ln\left[\left(\frac{1}{1-\sigma }-\frac{1}{1-R}*\frac{\left(1-\sigma \right)}{\sigma }\right)\right]=-\frac{\left(1-\sigma \right)}{P_{S}}*J_{v}$$

The measured flux and experimental rejection results were fitted into the SKK, as shown in Equation (7). A python program was employed to solve the non-linear equation through fitting. Then, the two unknown parameters, σ and $P_{S}$, were computed using a least-squares minimization.

## 3. Results and Discussion

Orange II, riboflavin and naphthol green B (Figure 1) were used as model solutes to investigate the mass transport behavior through the prepared TFC membranes, M1 and M2. Orange II and naphthol green B acquire negative charges when dissolved in water, while riboflavin remains uncharged. The surface and cross-sectional morphologies of the prepared M1 and M2 TFC membranes were investigated using SEM (Figure 2). Both M1 and M2 had porous surfaces (Figure 2a,b). The cross-sectional SEM images (Figure 2c,d) demonstrate that both M1 and M2 membranes possessed a thin TA–Fe^3+^ skin layer deposited on top of the porous PAN support. The PAN support had a spongy integral asymmetric structure. For both M1 and M2, the ultrathin TA–Fe^3+^ layer only existed at the top of the membranes and the spongy porous structure of the PAN support was not blocked at all. In general, for integral asymmetric porous membranes, the resistance against the mass transport of the permeating substance decreased from the top towards the bottom as the pore size gradually increased along the cross-section of the membrane. Figure 3a shows that the water flux through the PAN support was more than ten times higher compared to that through M1 and M2. The resistance against mass transport of the permeating substances through M1 and M2 increased significantly due to the deposition of the TA–Fe^3+^ layer on the PAN. Hence, the retention and flux through the M1 and M2 membranes were solely dictated by the TA–Fe^3+^ and the contribution of the porous PAN substructure was negligible. The top surface images of both M1 and M2 membranes (Figure 2a,b) show small pores unevenly distributed over the surface. Comparatively, the membrane prepared at higher pH (M2) had more closed pores. This was due to the fact that at a pH of 7 or higher, the complexation state of metals and polyphenols became a tris-complex, where three TA molecules coordinated with an iron center, leading to the formation of a compact layer [32,33,34]. It was not possible to decipher any substantial difference between the two membranes in terms of average surface pore sizes from the SEM images. The SEM image depicts only a very small area of the membrane and it is visible that the pore size distribution at the surface of M1 and M2 was rather broad. Figure 3b–d show the fluxes of riboflavin, orange II and naphthol green B aqueous solutions against the transmembrane pressure. While the fluxes of the aqueous solutions gradually increased with the transmembrane pressure in all cases, small deviations from the linear behavior were also observed. However, it is clear that the fluxes of the three aqueous solutions through M1 increased more sharply with transmembrane pressure compared to M2. Hence, it is evident that the average surface pore size of M1 was higher than that of M2. Photographs of the feed and permeate solutions investigated for M2 at a transmembrane pressure of 3 bar are shown in Appendix A.

The fluxes of the aqueous solutions over a transmembrane pressure range of 1–4 bar are plotted against the retention of the solutes (i.e., riboflavin, orange II and naphthol green B) by M1 and M2 in Figure 4a and Figure 5a, respectively. These experimental data points were used to determine the reflection coefficient, σ, and the permeability of the solute, $P_{S}$, using the non-linear fitting of the SKK model (Table 1 and Table 2). The obtained values of σ and $P_{S}$ were used as input parameters in Equation (7). Consequently, the non-linear relationship between the flux and retention of the solutes predicted by the SKK model was obtained for M1 and M2. The model algorithm used to estimate the parameters of the model is shown in Appendix A. The statistical procedure followed to fit the SKK model comprised the minimization of the residuals or error of the optimized parameters, which was a function of both the $P_{S}$ and σ model parameters. According to the SKK model prediction, with an increase in the flux of the aqueous solution, the retention of the solutes increased asymptotically (Figure 4a and Figure 5a). In order to check the validity of this prediction below the flux limit 5 g·m^−2^·s^−1^ (i.e., 5 × 10^−3^ kg·m^−2^·s^−1^), the two experimentally observed solution flux vs. solute retention points for each of the solutes were compared with the SKK predictions for M1 (Figure 4b) and M2 (Figure 5b). The experimental data points of the aqueous solutions of riboflavin showed a similar trend to the SKK model predictions for both M1 and M2. Only small deviations from the predicted values were observed. The experimental data points for orange II and naphthol green B solutions deviated substantially from the SKK model predictions for M1 (Figure 4b). Compared to that, smaller deviations between SKK model prediction and experimental data points for orange II and naphthol green B solutions were observed in the case of M2 (Figure 5b).

The asymptotic increase of the retention of an uncharged solute with the increase of solution flux as a result of increasing transmembrane pressure implied that there was competition between the convective transport of water and the solutes through the selective layer of the membrane. A low transmembrane pressure means a low driving force for the convective transport of molecules through a membrane. The driving force of convection increased under a high transmembrane pressure. On the other hand, the diffusive transport reached a limiting value [35]. The permeate volumetric flux was directly proportional to the applied transmembrane pressure, while solute permeability was not. The contribution of diffusion to solute permeability declined at a high permeate volumetric flux. Moreover, when the applied pressure was increased, water and solute fluxes became uncoupled [36,37]. The porous selective layer of membranes necessarily allows the transport of the molecules based on their size. Consequently, there was a selective transport of water molecules through the selective layer of the membranes. A large number of solute molecules bounced back from the surface of the membrane or were by the membrane due to their larger size compared to the water. The fluxes of the aqueous solution increased with transmembrane pressure due to a higher driving force (Figure 3b–d). It is essential to realize that, with the increase of transmembrane pressure, not only did a higher number of water and solute molecules move through the selective layer of the membrane per unit time, but the number of water and solute molecules that bounced back from the surface of the membranes per unit time also increased. Due to their smaller size, the water molecules had a higher probability of entering the pores of the selective layer compared to molecules of a larger uncharged solute. The asymptotically increasing solution flux vs. solute retention curves implies that there was a limiting value up to which the increase of transmembrane pressure resulted in a greater increase of the water flux compared to the solute flux. It is obvious that the limiting value of the transmembrane pressure where the solute retention vs. solution flux curve levels off would change depending on the size of the pores and solutes. The SKK model is a phenomenological model and the sizes of the pores and solutes are not used as input parameters. However, the reflection coefficient, σ, and the permeability of the solute, $P_{S}$, were calculated by fitting the experimental data points in a solution flux vs. solute rejection plot. The values of these model parameters inherently reflected the impact of the pore and solute size on the solution flux vs. solute retention curves predicted by the SKK model.

The values of σ followed the order of riboflavin < orange II < naphthol green B. Although riboflavin and orange II had comparable sizes, the flux of orange II through the membranes was substantially lower than that of riboflavin. Hence, it is evident that the retention of orange II by M1 and M2 was not only a result of size exclusion. To analyze the permeation and retention of the charged solute, the influence of Donnan exclusion must also be taken into account. The 0.1 mM orange II and naphthol green B dye solutions had a pH of 6.4 and 6.5, respectively [15]. At these pH values, TA–Fe^3+^ membranes are negatively charged, with an isoelectric point of less than pH 3 [31,38]. Our membranes also showed similar negative surface charges. M1 and M2 exhibited a zeta potential of −21.0 mV and −27.3 mV at pH 6.4, respectively. At the pH of these dye solutions, the negatively charged surfaces of M1 and M2 repelled the anionic dyes, resulting in a high rejection. Hence, the charged solutes (i.e., orange II and naphthol green B) had to overcome an additional energy barrier due to the charge repulsion (i.e., Donnan exclusion) compared to the uncharged solute (i.e., riboflavin). Since the driving force for convective flow was rather low at a very low transmembrane pressure, Donnan exclusion is likely to play a stronger role in the retention of the solutes. As the transmembrane pressure increased, the driving force for convective transport of the charged solutes increased as well. This means that the charged solutes had a higher probability of overcoming the additional energy barrier of Donnan exclusion and permeate through the pores at higher transmembrane pressure. Hence, there are two competing phenomena which finally determined the influence of transmembrane pressure on the retention of a charged solute. First, similar to uncharged solutes, charged solutes also had to compete with the solvent to enter pores of the membrane, which tended to increase the solute retention with increasing transmembrane pressure. Second, the charged solutes had a higher probability of overcoming the additional energy barrier of Donnan exclusion to enter the pores with increasing transmembrane pressure, which tended to decrease the retention. The SKK model only predicted the influence of the first phenomenon. In the case of a charged solute, σ and $P_{S}$ inherently reflected the influence of both size and Donnan exclusion, as they were calculated by fitting the experimental data points of the solution flux vs. solute rejection plot. However, as the SKK model failed to predict changes in retention due to the second phenomenon described above, a strong deviation of the retention behavior predicted by the SKK model from the experimentally obtained data was observed at low transmembrane pressures (Figure 4b and Figure 5b). According to our observation, at a low transmembrane pressure the second phenomenon had a stronger influence on the retention of orange II and naphthol green B by M1 (i.e., the membrane with a larger pore size) than M2.

It should be noted that the solute rejection increased until it stabilized when the applied pressure was increased for both M1 and M2 membranes (Figure 4a and Figure 5a). This phenomenon was shown by orange II and riboflavin. However, the rejection of naphthol green B slightly increased in M1 and decreased in M2 with an increase of fluxes.

Aside from pressure, the feed concentration played a vital role in membrane filtration performance. An increase of the concentration of feed solution increased the driving force for solute diffusion across the membrane. Therefore, the solute flux through the membrane was expected to rise, i.e., the solute rejection was expected to decrease. On the other hand, at a constant feed pressure, a high feed concentration of solute was expected to result in a lower permeate volume flux due to an increase in the osmotic pressure. In order to investigate the influence of a high concentration, we varied the concentration of the orange II and naphthol green B feed solution in the range of 0.01 mM to 1 mM. Although the feed concentration was varied 100 fold, the volumetric fluxes through M1 and M2 did not vary substantially (e.g., standard deviations of 2.9 and 2.7 g·m^−2^·s^−1^ (i.e., 2.9 and 2.7 kg·m^−2^·s^−1^) for M1 were observed for Orange II and naphthol green B, respectively). Orange II and naphthol green B were used in this experiment while riboflavin was dropped due to its low solubility in water. Figure 6a,b present the real rejection of dyes and model-predicted results as a function of the feed concentration for both membranes. The observed rejection decreased slightly with an increase in the feed concentration in all case studies. For instance, the naphthol green B rejection declined from 98.7 to 96.3% in M1, while orange II rejection decreased from 94.0 to 90.0% in M2. However, as the concentration exceeded 0.05 mM, both membranes showed an almost constant performance with a limiting value of rejection. Hence, these results show that the diffusive transport of dyes through M1 and M2 was largely prevented owing to the size and charge exclusion mechanism. As a result, even a 100-fold change in the feed concentration did not have a substantial influence on the separation performance of the membranes. The SKK model also predicted a similar trend. A slight variation between the experimental and predicted results was observed at low feed concentrations for all dyes in both membranes. Rejection was underestimated by the SKK model. This difference was due to the fact that the driving force of diffusion (i.e., the concentration gradient) was not an input parameter in the model. The rejection was predicted from Equations (5) and (6) using the volumetric fluxes, reflection coefficients and permeability of the solutes as input parameters. The simulation was performed using the reflection coefficient and solute permeability parameters obtained in Table 1 and Table 2. At a transmembrane pressure of 3 bar, the volumetric flux remained unchanged at feed concentrations of 0.01–1 mM, while the predicted rejection of the dyes was also constant. The limitation of the SKK model to adequately describe membrane performance of dilute feed systems has been reported in other studies as well [21].

Figure 7 presents the SKK model-predicted dye rejection as a function of their experimentally determined rejection values for all case studies. It also includes comparisons of rejection at different feed concentrations. An even distribution of many data points along and close to the diagonal line demonstrated the significance of the model. In accordance with these results, the applied model fitted the experimental data well. This study shows that the TA–Fe^3+^ membrane performance can be predicted using SKK model parameters and experimentally observed volumetric fluxes under the operating conditions where convection is the dominant transport mechanism.

## 4. Conclusions

The prediction of dye rejection performance of metal–polyphenol based membranes using the Spiegler–Kedem–Katchalsky model was investigated in this work. The membranes exhibited a high rejection towards negatively charged dyes; the reflection coefficient, *σ*, was in the range of 0.90–0.96 for anionic dyes of orange II and naphthol green B, while *σ* values of 0.15 and 0.18 were found for uncharged riboflavin. An analysis of the phenomenological model for dye transport successfully illustrated how permeate flux affects solute rejection. Experimental and predicted values were in good agreement. The validity of the model was also evaluated at low transmembrane pressure as well as using different feed concentrations. Operations at low transmembrane pressure showed a deviation between model-estimated and observed data due to the strong influence of Donnan exclusion. The volumetric fluxes through the membranes at a transmembrane pressure of 3 bar did not change significantly even with a 100-fold change in the feed concentration owing to the influence of size and Donnan exclusion. Consequently, the values of dye retention predicted by the SKK model at diluted feed concentrations deviated from the empirical values. While the overall model outcomes matched experimental data to some extent, some major limitations of the SKK model to predict the retention of charged dyes by the TFC membranes with TA–Fe^3+^ selective layers were shown in this study. These conclusions are likely to be true for other membranes as well.

## Figures and Tables

**Figure 1 membranes-12-01216-f001:**
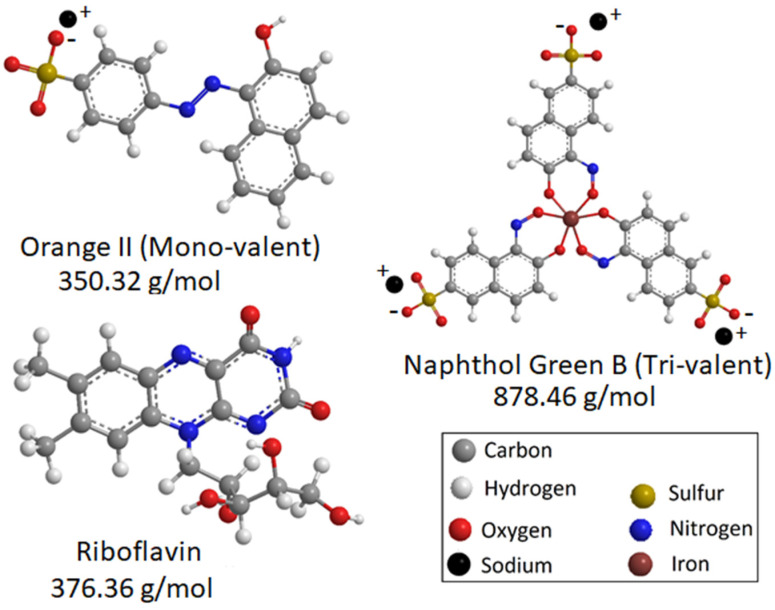
Chemical structure and molecular weight of the orange II, riboflavin and naphthol green B solutes.

**Figure 2 membranes-12-01216-f002:**
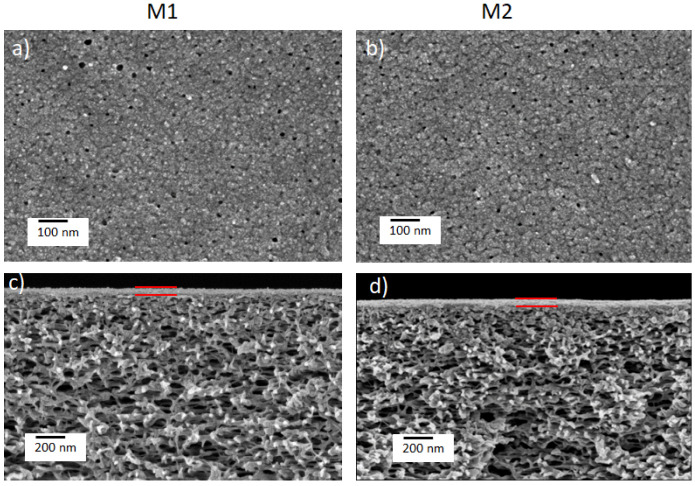
SEM images of top surface and cross-section (**a**,**c**) of M1 and (**b**,**d**) M2 thin-film composite membranes. In the cross-section images (**c**,**d**), the TA–Fe^3+^ selective layers are pointed out with two red lines.

**Figure 3 membranes-12-01216-f003:**
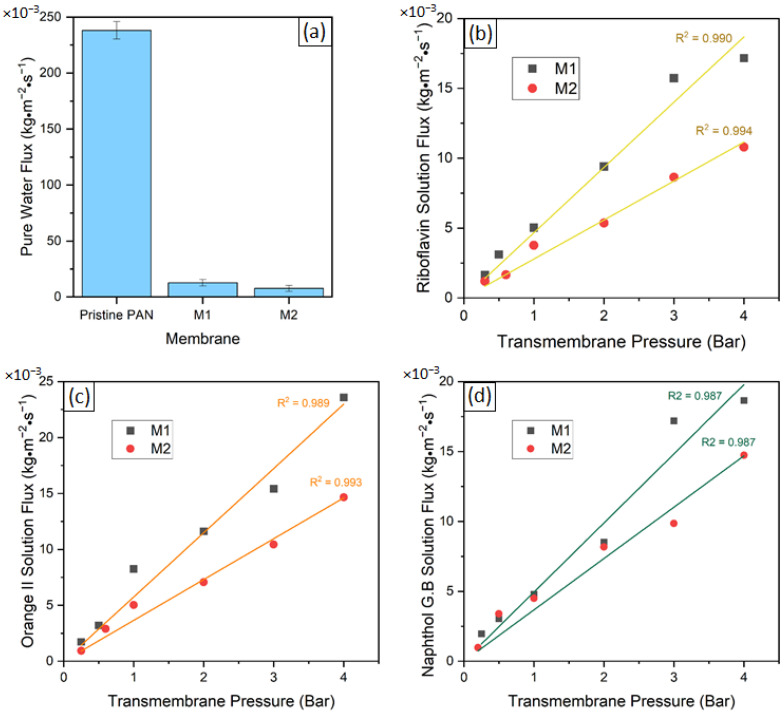
(**a**) Pure water flux at a pressure of 3 bar for pristine PAN support, M1 and M2 membranes; and solution flux as a function of transmembrane applied pressure for (**b**) riboflavin, (**c**) orange II and (**d**) naphthol green B.

**Figure 4 membranes-12-01216-f004:**
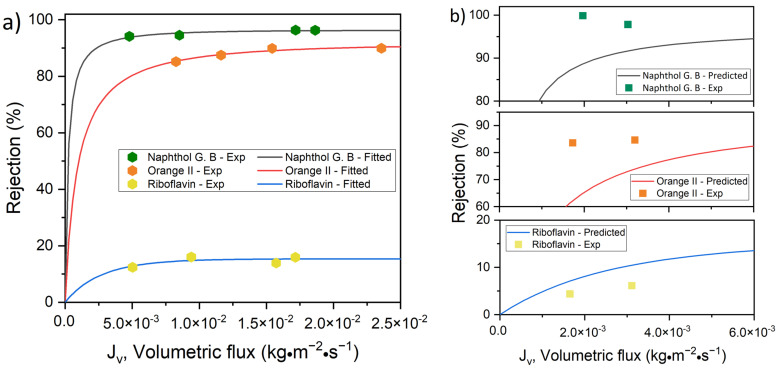
Dye retention versus flux for M1 (1 TA–4.5 Fe, pH 5.8) membrane; (**a**) experimental results and curve-fitting for SKK model and (**b**) validation of model through low flux experimental rejection.

**Figure 5 membranes-12-01216-f005:**
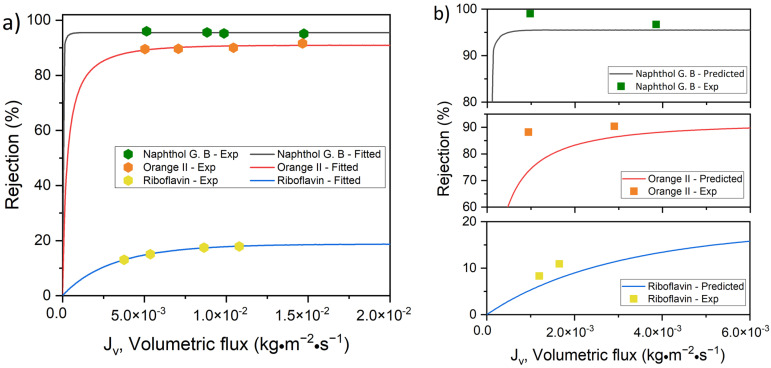
Dye retention versus flux for M2 (1 TA–4.5 Fe, pH 8.5) membrane; (**a**) experimental results and curve-fitting for SKK model and (**b**) validation of model through low flux experimental rejection.

**Figure 6 membranes-12-01216-f006:**
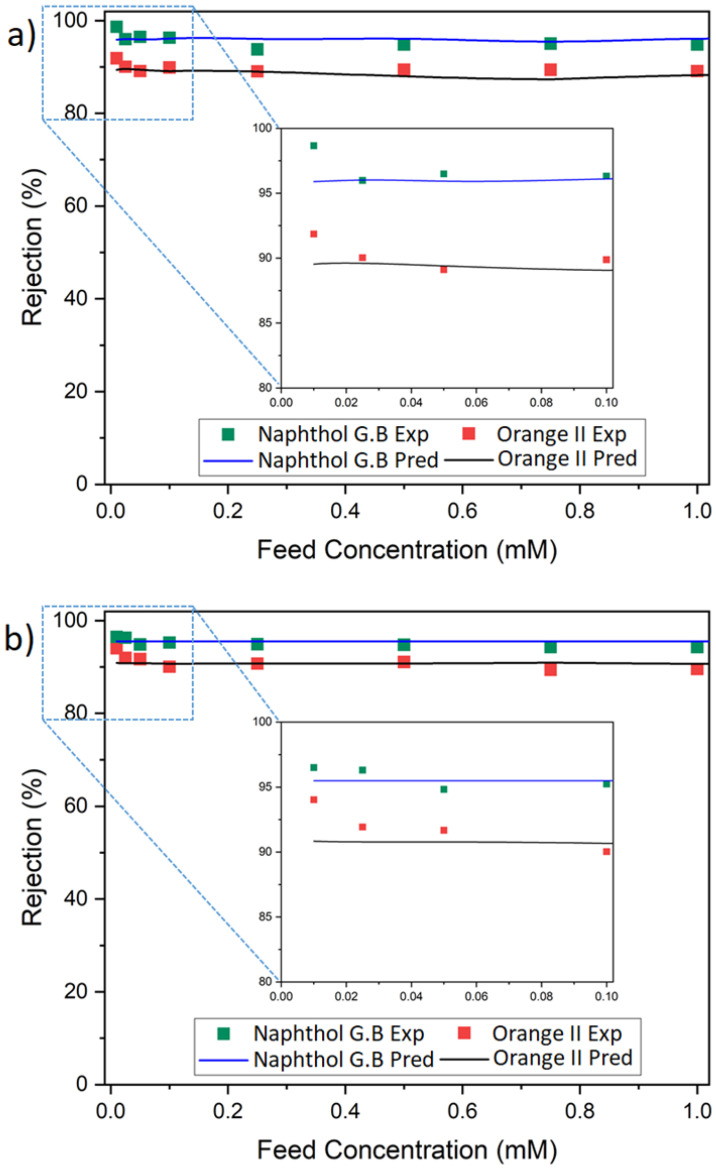
Dye retention prediction at different feed concentrations for 1 TA–4.5 Fe membranes prepared with (**a**) DI water of pH ∼ 5.8 (M1) and (**b**) at pH 8.5 (M2). Exp denotes experimental data, while Pred implies prediction line.

**Figure 7 membranes-12-01216-f007:**
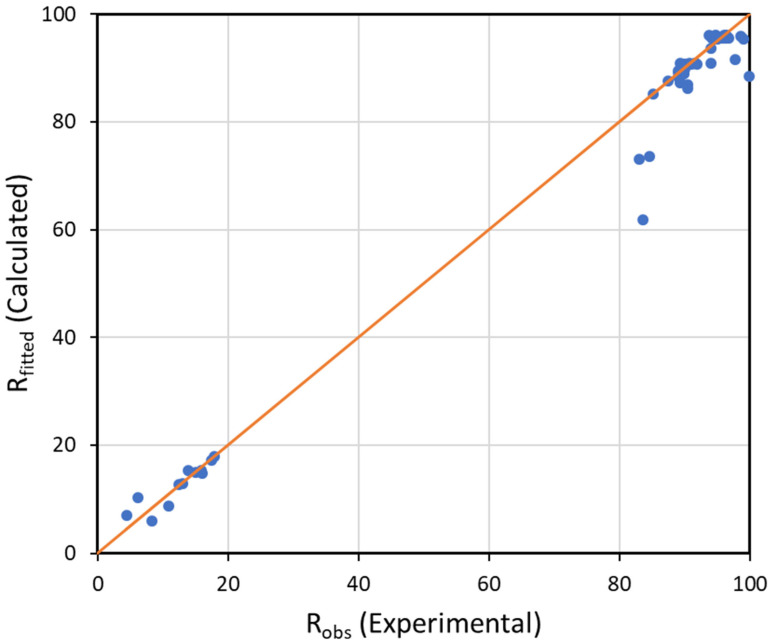
Comparison of experimental and model-predicted rejection for all measurements in this work.

**Table 1 membranes-12-01216-t001:** Model transfer coefficient for membrane M1.

Dye	*σ*	*P_S_* (m/s)
Riboflavin	0.153822	2.57 × 10^−6^
Orange II	0.911965	8.83 × 10^−7^
Naphthol Green B	0.962552	2.04 × 10^−7^

**Table 2 membranes-12-01216-t002:** Model transfer coefficient for membrane M2.

Dye	*σ*	*P_S_* (m/s)
Riboflavin	0.187374	2.92 × 10^−6^
Orange II	0.909181	2.65 × 10^−7^
Naphthol Green B	0.954968	1.04 × 10^−8^

## Data Availability

Not applicable.

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
