# Peer review of "Mass Transport of Dye Solutions through Porous Membrane Containing Tannic Acid/Fe3+ Selective Layer"

_membranes, 2022, doi:10.3390/membranes12121216_

Round 1

Reviewer 1 Report

It is of great significance to accurately predict the influence of various factors on the mass transfer behavior in membrane filtration process for developing high-quality membranes and optimizing filtration process. In this work, the authors used SKK model to predict the retention of dyes in porous film composite membranes containing metal polyphenol networks (MPNs) selection layers. This work is a meaningful attempt to deepen membrane scientists' understanding regarding the correlations between the nature of solutes, membrane morphology and modes of transport. Considering the originality and importance of this work, I recommend it for publication in Membranes after addressing the following issues:

1. In this manuscript, I noticed that the dye rejection rate (R) was calculated using the following equation:

But as far as I know, the retention rate (R) of membrane is generally calculated by the following equation:

R (%) = (1-Cp/Cf) × 100     (1)

Note: where Cf and Cp are the concentration of feed and permeate, respectively.

Comparing the above two equations, we can clearly see that the authors use (Cf+Cr)/2 instead of Cf to calculate the interception rate. Why?

2. Note that there are spelling errors in the text, for example, "semipearmeable" should be "semipermeable".

3. In the morphological characterization of TFC membranes, the authors should clearly mark the TA-Fe3+ layer and its thickness in cross-section SEM images of M1 and M2 membranes.

4. I was confused by the content in the draft describing the change of the membrane's rejection rate of charged molecules with transmembrane pressure: "Second, unlike uncharged molecules the charged solutes have larger probability to overcome the additional energy barrier of donnan exclusion to enter the pores with increasing transmembrane pressure which will tend to decrease the retention." Why are charged molecules more likely than uncharged molecules to overcome the donnan exclusion and enter the membrane pores?

5. Some reported literatures on the influence of charge properties of membranes on the retention performance of different charged molecules should be referred (Journal of Materials Chemistry A, 2018, 6 (27), 13331-13339. 2200718. Separation and Purification Technology, 2021, 256, 117787. Macromolecular Rapid Communications, 2022).

Author Response

Reviewer 1.

Recommendation: Publish after minor revisions noted.

Comments: It is of great significance to accurately predict the influence of various factors on the mass transfer behavior in membrane filtration process for developing high-quality membranes and optimizing filtration process. In this work, the authors used SKK model to predict the retention of dyes in porous film composite membranes containing metal polyphenol networks (MPNs) selection layers. This work is a meaningful attempt to deepen membrane scientists' understanding regarding the correlations between the nature of solutes, membrane morphology and modes of transport. Considering the originality and importance of this work, I recommend it for publication in Membranes after addressing the following issues:

  1. In this manuscript, I noticed that the dye rejection rate (R) was calculated using the following equation:

But as far as I know, the retention rate (R) of membrane is generally calculated by the following equation:

R (%) = (1-Cp/Cf) × 100              (1)

Note: where Cf and Cp are the concentration of feed and permeate, respectively.

Comparing the above two equations, we can clearly see that the authors use (Cf+Cr)/2 instead of Cf to calculate the interception rate. Why?

Authors’ response- In membrane processes, especially in cross-flow configuration, the rejection equation provided by the reviewer is generally used. However, for dead-end cell filtration configuration, the feed concentration above the membrane surface is constantly changing with time as permeation proceeds. To minimize the change of feed concentration we used a large feed volume and a rather small membrane surface area. Inspite of that we can’t deny the feed concentration may change slightly with the passage of time. Two take it into account we check the concentration of at the beginning of permeate collection Cf and at the end of permeate collection Cr. In this way (Cf+Cr)/2  represent the average concentration during the permeate collection. In short, we use (Cf+Cr)/2  instead of Cf for determination of R to take into account the minor change of concentration during the experiment. 

  1. Note that there are spelling errors in the text, for example, "semipearmeable" should be "semipermeable".

Authors’ response- Spelling errors are fixed now.

  1. In the morphological characterization of TFC membranes, the authors should clearly mark the TA-Fe3+ layer and its thickness in cross-section SEM images of M1 and M2 membranes.

Authors’ response- Thickness of the TA-Fe3+ layer is very thin that it was difficult to measure it with the used SEM device’s resolution (few nanometers). The synthesized dense layer is shown in between two newly added red lines.

  1. I was confused by the content in the draft describing the change of the membrane's rejection rate of charged molecules with transmembrane pressure: "Second, unlike uncharged molecules the charged solutes have larger probability to overcome the additional energy barrier of donnan exclusion to enter the pores with increasing transmembrane pressure which will tend to decrease the retention." Why are charged molecules more likely than uncharged molecules to overcome the donnan exclusion and enter the membrane pores?

Authors’ response- This sentence explains how charged solutes behave at different applied pressures with respect to the energy barrier for overcoming donnan exclusion. The comparison is between a charged solute at low and high flux. But, the “unlike uncharged molecules” is stating that neutral solutes do not exhibit such kind of energy barrier variation irrespective of pressure difference because they do not have donnan exclusion mechanism. To avoid confusion we deleted “unlike uncharged molecules” from the above sentence (page 11, lines 301 and 302) as it is clear from the context.

  1. Some reported literatures on the influence of charge properties of membranes on the retention performance of different charged molecules should be referred (Journal of Materials Chemistry A, 2018, 6 (27), 13331-13339. 2200718. Separation and Purification Technology, 2021, 256, 117787. Macromolecular Rapid Communications, 2022).

Authors’ response- Journal of Materials Chemistry A, 2018, 6 (27), 13331-13339.  Separation and Purification Technology, 2021, 256, 117787 and Macromolecular Rapid Communications, 2022, 2200718 are cited in the revised manuscript.

Reviewer 2 Report

In this work, the Spiegler-Kedem-Katchalsky model was used to calculate the transport of various dye solution using an active separation layer of tannic acid/Fe3+ on top of PAN macroporous support. The membranes were adequately characterized and the idea was clearly proven through supported discussions and experimentation. In my view, the paper can be accepted after the following minor revision:

1.      The membrane’s surface is charged. Did the author(s) use positively charged dye solution for transport calculations?

2.      The molecular weight between the two anionic dyes varies considerably (350 g/mol for Orange II vs 878 g/mol for Naphthol Green B). Did the author(s) consider any anionic dyes with molecular weights above 878, below 350 or in between these values?

3.      Some minor spell mistakes throughout manuscript especially the term Naphthol, such as Naphtol instead of Naphthol in Fig.1; napthol instead of naphthol in Fig.1 caption etc.

Author Response

Reviewer 2.

In this work, the Spiegler-Kedem-Katchalsky model was used to calculate the transport of various dye solution using an active separation layer of tannic acid/Fe3+ on top of PAN macroporous support. The membranes were adequately characterized and the idea was clearly proven through supported discussions and experimentation. In my view, the paper can be accepted after the following minor revision:

  1. The membrane’s surface is charged. Did the author(s) use positively charged dye solution for transport calculations?

Authors’ response- We have observed that positively charged dyes (e.g. Rhodamine) show strong affinity for adsorption on the membrane surface due to electrostatic interactions. As SKK model is not the right model to study the removal of dyes via adsorption such study is not relevant to this paper.

  1. The molecular weight between the two anionic dyes varies considerably (≈350 g/mol for Orange II vs ≈878 g/mol for Naphthol Green B). Did the author(s) consider any anionic dyes with molecular weights above 878, below 350 or in between these values?

Authors’ response- No, we have not tried other anionic dyes below or above the mentioned molecular weight.

  1. Some minor spell mistakes throughout manuscript especially the term Naphthol, such as Naphtol instead of Naphthol in Fig.1; napthol instead of naphthol in Fig.1 caption etc.

Authors’ response- Spelling errors in the main manuscript and supplementary information are fixed now. Figure 3, 4, 5 and 6 are also edited for a name change of ‘naphthol’ in their legends/axis.